# Impact of Physical Activity on Anxiety, Depression, Stress and Quality of Life of the Older People in Brazil

**DOI:** 10.3390/ijerph20021127

**Published:** 2023-01-08

**Authors:** Helena Andrade Figueira, Olivia Andrade Figueira, Alan Andrade Figueira, Joana Andrade Figueira, Reinaldo Emilio Polo-Ledesma, Carlos Roberto Lyra da Silva, Estelio Henrique Martin Dantas

**Affiliations:** 1Programa de Pos-Graducao em Enfermagem e Biociencias (PPgEnfBio), Universidade Federal do Estado do Rio de Janeiro (UNIRIO), Rio de Janeiro 22290-180, Brazil; 2Escola de Ciencias da Vida do Programa de Pos-Graduacao em Bioetica (PPgB), Pontificia Universidade Catolica do Parana (PUR-PR), Curitiba 80215-901, Brazil; 3Faculdad de Ciencias de La Salud, Universidad SurColombiana, Neiva 30933-85, Colombia

**Keywords:** physical activity, anxiety, depression, stress, quality of life, older people

## Abstract

Background: Population aging all over the world invites older people to be active, considering physical activity (PA) as associated with reduced anxiety, depression and stress and a high quality of life (QOL) in older people. Objective: To analyze anxiety, depression, stress and QOL as a function of PA in older people. Methodology: Descriptive analytical research. Six hundred and ninety older people answered the instrument composed of sociodemographic questions, followed by questions from the WHOQOL-Old, Baecke-Old, WHOQOL-SRPB, Stress Perception Scale, Beck Anxiety Inventory and Beck Depression Inventory. Results: The beneficial effect of PA on the elderly is evident in this sample with such a high QoL (73%) and such a high level of physical activity (84%), and even with an advanced level of education (75%) and high spirituality (99.6%). The influence of PA on the anxiety and stress cluster showed Pearson’s chi-square = 9.9, DF = 4, *p* = 0.04239, critical value = 9.5. The influence of PA on the anxiety, depression and stress cluster showed Pearson’s chi-square = 6.8; DF = 5; *p*-value = 0.24; critical value = 11.1. Conclusion: In the elderly, PA has a significant relationship with anxiety, stress and QOL. In addition, the high level of QOL of the elderly in the sample demonstrates the capacity for PA, educational level and spirituality for personal satisfaction.

## 1. Introduction

Population aging [1] has become the greatest challenge of the 21st century [2], and indicates that by 2050 the world’s older population, individuals aged 60 years and over [3], will have a ratio of one older person for every five people [4]. Worldwide, nations are developing and implementing healthy aging policies, promoting the practice of physical activity (PA), and understanding sedentary lifestyle as a risk factor for chronic and degenerative diseases, the fourth risk factor for mortality globally [4]. The aging process and low levels of physical conditioning are associated with progressive reductions in functional capacity, increased fat and progressive muscle decline [5], the quality of life (QOL) and functional autonomy [6]. A non active lifestyle is associated with higher levels of anxiety and depression, and reduced mental health and the QOL [7]. The QOL is a subjective and broad concept with multiple dimensions that encompass physical, psychological, emotional, mental, spiritual, social and environmental dimensions [8]. Defined as subjective well-being reflecting the distance between one’s own goals, expectations, standards and concerns and the actual experience, supported by the achievement of goals according to the value system in which the person is inserted [9]. There is consensus in the literature about the positive effects of PA on the QOL [10], providing health and well-being [11]. Associated with the reduction of the QOL is depression in the elderly, which favors social isolation and the emergence of severe cases of clinical diseases [12]. Approximately one in five older people may suffer from depression, the second most common cause of physical and psychological disability in older people, with significant social and occupational impairment [13]. Being involved in a PA is enough to act as a protective factor against depression, regardless of frequency and duration [11]. 

Often associated with depression, anxiety [14] is the most prevalent psychiatric disorder, and is related to a high burden of disease [15]. Depression and anxiety are still considered diseases of old age, having important effects on the QOL, mortality and morbidity of the older people [15]. Sedentary older people have more anxiety, depression and stress than those who exercise [16]. PA is a protective factor against stress, with no significant difference by PA intensity [17]. Accelerated contemporary development, with a tendency to wear out physical and mental integrity, made stress the main disease of the century; the “epidemic” of the 21st century [18]. Stress designates an emotional arousal and an adaptation process, a general and non-specific response of the organism to a stressor, however a neglected public policy goal [17]. Chronic exposure to stress with prolonged exposure to cortisol can lead to changes in mental health [19]. Spirituality is considered a protective factor for anxiety, depression and stress [20]; anxiety and depression are affected by spirituality and PA in aging [13]. 

The older person is a bio-psycho-socio-spiritual being with multiple dimensions that constitute their integrality [21]. Active aging appears as an appeal to a pattern that sees the elderly as a participant in a society that integrates them, both as a contributor and a beneficiary of development [22]. Old age enhances limitations, and therefore much is required from an adaptive point of view, and non-adaptation can generate an imbalance [21]. Policies and programs which stuck to the outdated paradigm that associates aging with disease and dependence, do not reflect reality, as older people often remain independent even at a very advanced age, adhering to programs promoting their health [23]. 

This study is an analysis of the effect of PA on the aging process based on the following variables: anxiety, depression, stress and the QOL of the older person. The objective of the present study was to analyze the influence of PA on anxiety, depression, stress and the QOL of older people. The research hypothesis was that older people find, in the practice of PA, an alternative to feeling less anxious, less stressed and less depressed, thus achieving a higher level of the QOL. The PICO strategy was used to elaborate the guiding question of this research being P- Population: older people; I- Intervention: regular practice of PA; C- control group: no regular practice of PA; O- Outcome: anxiety, depression, stress and QOL. 

## 2. Materials and Methods

### 2.1. Sample and Procedure

Cross-sectional, descriptive, analytical observational research and ex-post-facto survey in compliance with the Declaration of Helsinki (18), approved by the Research Ethics Council—UNIRIO # 3,670,727.

The sample consisted of 690 community-dwelling older people, of both genders, unselected, volunteers, Brazilians, presents (as a participant or observer) in urban street races in Rio de Janeiro, Brazil, from 30 October 2019 to 12 March 2020: a. Búzios 11 October 2019; Rio Claro 17 November 2019; Resende 17 November 2019; Barra Mansa 30 November 2019; Resende 8 December 2019; Paraty 15 December 2019; Resende 22 December 2019; Rio de Janeiro 25 January 2020; Rio de Janeiro 26 January 2020; Rio de Janeiro 16 February 2020; Barra do Pirai 8 March 2020; Rio de Janeiro 8 March 2020. The exclusion criteria were blindness, deafness or cognitive impairment, which would make it difficult to have a clear idea when answering the questionnaire.

### 2.2. Measures

The self-administered survey instrument with 43 questions, 10 of which were sociodemographic, followed by questions selected from validated inventories, answered on a Likert scale: seven from the World Health Organization Quality of Life inventory for older people, WHOQOL-Old, five from the Baecke-Old, four from WHOQOL-SRPB, six from the Perception Stress Scale, PSS, six from the Beck Anxiety Inventory, BAI, and five from the Beck Depression Inventory, BDI-II-II. The WHO’s World Health Organization Quality of Life group developed, worldwide, a specific QOL instrument for the older people, in a cross-cultural perspective, the WHOQOL-OLD, translating physical, emotional and social health status [4]; and a World Health Organization Quality of Life inventory for affirming personal, religious and spiritual beliefs, WHOQOL-SRPB [24]. WHOQOL-Old [4] questions with responses on the Likert scale from 0 to 4: a. sense losses affect your daily life; b. how much freedom you have to make your own decisions; c. you are afraid of dying; d. you get to do what you would like to do; e. you are satisfied with opportunities to continue achieving achievements in your life; f. you feel you have received the recognition you deserve in your life; g. you feel love in your life. WHOQOL-SRPB [24]. Questions: your personal beliefs give meaning to your life; you feel inner peace; you consider yourself a religious person; the extent to which you are part of a religious community. The internationally validated Baecke-Old inventory [7] measures PA in the older people in broad approach [25]. BAECKE-Old, with five questions with four alternatives on a Likert scale from 0 to 3: a. performs heavy housework, b. flights of stairs per day, c. means of transport, d. practice physical activity, e. in leisure you…; answers: 0—never, 1—sometimes; 2—almost always; 3—always. Classifying anxiety and depression into levels, Aaron Beck developed a transcultural perspective, inventories of anxiety and depression, BAI, Beck Anxiety Inventory [26], and BDI-II, Beck Depression Inventory–II [27]. Beck Anxiety Inventory, BAI, six questions: do you feel a. riled up; b. unable to relax; c. fear; d. palpitation or racing of the heart; e. nervous; f. suffocated. Likert responses from 0 to 3 (0 = not at all to 3 = extremely). Beck Depression Inventory, BDI-II, five questions: do you feel a. disappointed in yourself; b. less capable than the people around you; c. with thoughts of killing oneself; d. sadder than usual; and. failed. With four alternatives on a Likert scale from 0 to 3: absolutely not; slightly; moderately; seriously. The Perception Stress Scale, PSS, by Sheldon Cohen [28] is the most widely adopted instrument to measure the perception of feelings and thoughts of stress in the last month. Perceived Stress Scale, PSS, six questions: have you been feeling a. sad about something unexpected; b. that you are not able to control important things in your life; c. nervous and “stressed; d. unable to handle everything you need to do; e. able to control irritation in your life; f. with difficulties that pile up so high that you cannot overcome them. Answers from 0 to 4 on the Likert scale were: absolutely not; slightly; moderately; seriously. 

### 2.3. Statistical Methods

In its global recommendation on physical activity for health, the WHO includes recreational, occupational and leisure physical activities, transportation (walking or cycling), home care, games, sports and exercise in their physical activity, advising to maintain a minimum of 150 min of moderate-intensity aerobic exercise per week or 75 min of vigorous-intensity weekly exercise, and the practice of weight training involving the main muscle groups at least twice a week [3]. In compliance with the WHO guideline for physical activity in the older people, the gradation adopted in this research was from 0 to 3 classified as sedentary, and above 3 considered active. In evaluating the results, personal beliefs was answered by “your personal beliefs give meaning to your life”, religiosity was answered by “you consider yourself a religious person”, and spirituality was answered by “you feel inner peace”, according to the guiding definitions of the WHOQOL team that elaborated the SRPB. The SRPB domain second component represents spirituality and personal beliefs and encompasses the facets of inner peace, hope and optimism, and purpose in life. While the item does not explicitly show spirituality itself, the guiding definitions have included components of spirituality and, therefore, this meaning is embraced [24]. The total percentage score of responses of WHOQOL-Old was classified: 0–40 = low; 41–69 = moderate; 70–100 = high. The gradation levels of the Perceived Stress Scale [28] are: 0–8 low; 9–16 moderate; 17–24 perceived. The evaluation of scores suggested by Beck was weighted, resulting in the score categories: of BAI. 6–9 = without; b. 10–11 = mild; c.12–15 = moderate; d. 16–18 = severe [26]; of BDI: a. 0–3 = without; b. 4–5 = mild; c. 6–7 = moderate; d. 8–15 = severe [27]. 

Data were automatically transferred from the online questionnaire to MS Excel 2019, where they were initially processed. Subsequently, the data were transferred to Epi info 7 for advanced statistical analysis. 

Considering that the objective of the study was to analyze anxiety, depression, stress and QoL as a function of PA in the elderly, we created the clusters: ADS and QoL. these clusters allowed working with two dependent and one independent variables. The two dependent variables, the ADS and QL cluster, composed respectively of: by QL and the ADS cluster (=Anxiety, Depression and Stress), and the independent variable being PA. To elaborate the ADS and AS clusters, PSS is classified into three levels (no, mild to moderate and perceived); and BDI and BAI into four levels (no, mild, moderate and severe), so that the BDI and BAI clustered with the PSS, we leveled the BDI and BAI scales into three levels (joining their mild and moderate levels, as the rating of the PSS). The statistical analysis made two comparisons: one with the ADS cluster, including depression (ADS = anxiety, depression and stress) and another with the AS cluster (AS = anxiety and stress), without including depression.

## 3. Results

The average age of the sample was from 60 to 69 years old, 74% were female, 94% had a high school diploma, 75% had a university degree, 69% lived with their family, 55% were married. According to the WHO classification, the sample had 582 active older people, 84.35%, and 108 sedentary, 15.65%. SRPB, spirituality, religiosity and personal beliefs showed that for 99% of the elderly in the sample, personal beliefs give meaning to their lives, 99.6% feel inner peace, 93% are religious and 79% are part of a religious community. The sociodemographic profile of the older people sample is showed in Table 1.

The total QOL sample mean was 73% and the standard deviation was 13%; 27% no anxiety, 35% mild anxiety; depression 81% were without; 84% were under mild and moderate stress. Comparing active and sedentary older people: Pearson’s Chi-square anxiety = 3.5 > 0, *p*-value = 0.18, df = 2 and critical value = 5.99, the Chi-square lower than the critical value, the difference found is not significant; Pearson’s Chi-square depression = 1.5, *p*-value = 0.06, df = 3 and critical value = 1.18, Chi-square greater than the critical value, difference is significant, with 94% certainty; Pearson’s Chi-square stress = 0.41, *p*-value = 0.81, df = 2 and the critical value = 5.99, the Chi-square lower than the critical value, the difference found is not significant; QOL, Critical Pearson’s Chi-square = 19,218, *p*-value = 0.000067, df = 2, with 99.993% certainty, this conclusion observed in the sample can be applied to the elderly population in general, in society. 

The two dependent variables, the ADS and QOL clusters, composed respectively of: ADS = Anxiety, Depression and Stress and QOL, and the independent variable is AF made possible two comparisons, one with the cluster including depression (ADS = anxiety, depression and stress), and another without including depression in the AS cluster (AS = anxiety and stress). Table 2 presents the comparison with the cluster AS that had the following statistical results: Pearson’s chi-square = 9.89, *p* value = 0.042, df = 4 and critical value = 9.49, so anxiety and stress have a significant relationship with PA. 

Table 3 presents the cluster ADS that had the following statistical results: Pearson’s chi-square = 6.79; DF = 5; *p*-value = 0.24; critical value = 11.1, chi-square lower than the critical value, anxiety, depression and stress do not have a significant relationship with PA.

Analyzing the levels of anxiety, depression, stress and the QOL, according to PA, Figure 1 was obtained, representing a percentage of the value obtained from the data from the answers to the five questions, quantifying levels of anxiety, depression, stress and the QOL. The answers were leveled in null, mild, moderate and severe, according to the hierarchical model of the Likert scale. A non-parametric strategy, grouped the data by categories, in seven clusters, and Spearman’s correlation coefficients allowed to study the associations between the most relevant information crossings, according to the obtained scatterplot profiles and the respective Pearson regression coefficients. Figure 1 presents the vertical bar diagram of the Spearman regression coefficient for the studied combinations: AQ, AP, AD, AB, QP, QD, QB. The initials A, Q, P, D, B correspond to: A = Physical Activity, Q = Quality of Life, P = Psychic State (anxiety, stress and QOL), D = Depression, B = Well-being (anxiety, depression, stress and the QOL). Each combination of binary letters indicates the type of cross studied, for example, AQ looks for the association between Physical Activity and the Quality of Life, and QD between the Quality of Life and Depression.

Statistical regression analysis allowed to establish how the values of one variable depend on the data of another. The correlation coefficient, r, with values between −1 and 1, shows whether the correlation is inverse or direct, and the absolute value indicates how strong the association is. R values ranging from 0.3 to 0.7 indicate a moderate association and from 0.1 to 0.3 a weak correlation. In Figure 1, vertical bar diagram showing the values obtained for the Spearman regression coefficient for all combinations studied, it can be seen in each bar, the data for the crossings show a moderate association between the variables AQ = 0.69, AP = −0.57, AD = 0.46, BA = 0.38, and weak correlation between QP = −0.10, QD = 0.03, QB = 0.25. To confirm whether or not there is a correlation between all these selected combinations, Kendall’s coefficient of agreement was calculated, with which a value above 0.7 was obtained, indicating that the data obtained for the r values correspond to associations or divergences real values between the selected compared variables.

## 4. Discussion

The older people in this study live in the community and have a mean age of 60–69 years. They are characterized as PA practitioners (84%), female gender (74%), university level (75%), high QOL (73.35 ± 12, 6). Research with older people with lower educational and PA levels showed a lower QOL than in the present study: in older people, Iranians with a low level of education [29]; and in Brazil [1] and India [2] the QOL is low.

In the present research, the QOL stands out as remarkably high, in comparison with lower results than those found in Rio de Janeiro, Brazil, with physically active older people, QOL = 65.8% [30], in the Netherlands QOL = 60% [31], in Brazil, QOL between 50% and 72% [32].

PA is the lifestyle factor most strongly associated with frailty, especially physical frailty, both moderate or vigorous PA, as well as routine domestic PA, which are recognized as protective factors against frailty in older people [33]. Regular PA produces several benefits in older people; among others, a significant reduction in body mass index, a lower risk of falls, an increase in muscle mass and strength, flexibility, cardiorespiratory fitness, agility, dynamic balance improvement in the QOL allowing one to experience aging healthier [34]. The QOL is associated with the duration, frequency and intensity of the PA performed: the practice of PA for 120 to 180 min per week increases the QOL in relation to sedentary lifestyle, increasing the feeling of well-being, social relationships and social participation [4]. A study in Brazil with 1197 older people showed that the practice of PA is a factor that contributes to the QOL [35]. In a nationally representative sample of 1284 older people studied for 50 months in Taiwan, PA was a significant factor in independent living, an important domain of the QOL [36]. In Minas Gerais, research with 70 older people proved a positive relationship between PA and QOL [4]. A cross-sectional study with 200 older people of both genders, divided into two groups with equal numbers, between sedentary and PA practitioners, carried out in Cuiabá, Brazil, observed that the active group had higher QOL scores and the sedentary group, higher scores of anxiety and depression; revealing a strong correlation between the QOL domains, and the level of mental health, the prevalence ratio showed that PA is a protective factor against anxiety and depression in the older people [7]. The practice of PA showed a prevalence of no depression 87.5%, in a survey in Thailand, higher than in China 75.2% and lower than in India 91.8%, all confirming that a greater participation in PA increased the odds of successful aging, underlining the evidence that PA improves physical and mental health [37]. There is a close relationship between low QOL and high anxiety and depression [38]. PA suggested as a therapeutic proposal promotes consequent improvement in the QOL [39]. Physical activity and stress management are significantly associated with less physical, psychological and social frailty [33].

Spirituality, religiosity and personal beliefs are associated with a better QOL. The older people find in spirituality meaning for life and biopsychosocial comfort, mental health and the QOL, promoting social interaction, a positive attitude towards aging, acting as a health promoter [40]. Spirituality helps to promote the well-being of the older people; professional codes of ethics recognize the need for respect for beliefs and values associated with spirituality and its role in resilience and recovery from illness [41]. The results of this research with older people with spirituality (99.9%) with a high QOL (73.35%) confirm this statement. A high level of spirituality and contentment was found in a French study with 567 seniors of high academic level [42]. Greater personal spirituality is associated with a high level of education [43]. Healthy aging is associated with higher education [36]. The QQL proves to be quite low in older people with low education [29]. The present sample of high spirituality (99.5%) had a high educational level (75.4% with complete higher education) considering the Brazilian national average (17.4%), confirming a positive association between schooling and the QOL. Anxiety, depression and stress are significantly associated with the educational level of the older people [44]. 

There is growing recognition of the value of prevention programs for older people [45]. Frailty caused by advancing age is a major public health problem and a challenge for health care [33]. Physical activity (PA) is a behavioral complex that involves several areas beyond physical exercise, involving the structuring of a routine that can be performed collectively, contributing to socialization [3]. Lifestyles are referred to as conditioning factors for the frailty of older people [33]. Despite the proven benefits of PA, the sedentary lifestyle still tends to predominate in many countries. However, the effects of aging, such as muscle loss, which begins with reduced speed, strength, stability and firmness, are associated with serious health consequences. Delaying sarcopenia, muscle aging that compromises performance, increases firmness and stability, essential for individual and social well-being [46]. A public health policy agenda should include PA as a crucial element for healthy aging [47] benefiting older people, their direct environment and society at large [48]. Since PA is a promising non-pharmacological method in generating health, available to all people [49], the greatest possible investments must be made for its promotion. Policies and programs should encourage older people to become more active and should give them the opportunity to do so [22]. Older people should be as physically active as their abilities and conditions allow [4]. 

Aging presents challenges for the older people to reconcile with the past and reduce fears about the future, embracing this new phase of life. Aging with dignity is a choice and a goal, an individual and collective challenge, a moral and ethical proposal with rules of conduct and proposals for action [1]. Facilitating spirituality and PA can improve the QOL of the older people, alleviating anxiety, depression and stress. From the point of view of public health, it is essential to consider the factors that affect the QOL of the older people, with the intention of generating preventive and remedial strategies that can benefit the health of the population [34]. Promoting sustainable aging is a challenge by promoting new standards of individual and collective conduct, by promoting dialogue on values and objectives, based on scientific knowledge.

## 5. Conclusions

This study confirmed that anxiety, stress and the QOL have a significant relationship with PA in the elderly. In addition, the high level of QOL (73%) and PA (84%) of the elderly sample show the ability of a highly educated population (75% university level) with a high level of spirituality (99.6%) to foresee opportunities for personal satisfaction on the physical, psychic, social and spiritual levels.

The research carried out in the present study is essential to understand the peculiarity of the influence of PA on anxiety, depression, stress and the QOL of the elderly, considering that there is no published study associating these descriptors with each other. The complexity and heterogeneity of aging justify this multidimensional approach. Our findings reinforce the need to implement public policies aimed at the elderly, which consider health more broadly, knowing that several factors contribute to the development of health problems in the elderly. Health professionals and the entire community linked to them can help the elderly in a more adequate way, based on the theoretical and empirical approach of this study, whether in treatment, or in the elaboration of health policies and programs, of intervention strategies that stimulate the autonomy of the elderly, valuable for QOL, as well as talents to manage the aging process in all its nuances. The results found in this study can contribute to a better quality of life and physical and mental health of the elderly and to the development and improvement of public policies aimed at them.

### 5.1. Strenghts & Weakenesses

The present study has several strengths, including the sample size. Among its weaknesses is the spatial limitation to only one nation, without comparison with other cultural contexts; prospective research could cover other countries on other continents. Another limitation of this study is its cross-sectional design, which does not allow determining the action of time on the studied variables; ongoing prospective research may allow systematic analysis of PA and its relationship to anxiety, depression, stress, and the quality of life, as well as confounding factors. Other relevant limitations to be considered are that comorbidities, drug addiction, various aspects of functionality, and emotional and cognitive issues were not addressed. The multivariate forms of physical activity in their broad spectrum were not considered and may be an interesting study to be carried out in the future.

### 5.2. Difficulties 

The COVID-19 pandemic prevented the continuity of data collection, leading older people in Brazil to isolation at home on 12 March 2020. For this reason, the field research had to be closed with the sample obtained so far.

### 5.3. Suggestions 

The analysis carried out in this research did not make comparisons in the sociodemographic levels of marital status, residence, education, work and religion, which may be contemplated in future analyses. Nuances in the types of PA that increase the QOL of the elderly and decrease their anxiety, depression and stress can be contemplated in future studies.

## Figures and Tables

**Figure 1 ijerph-20-01127-f001:**
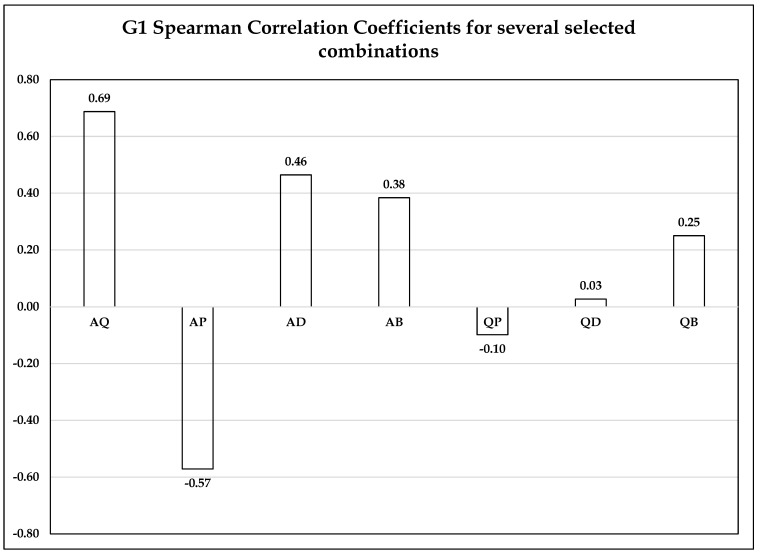
Vertical Bar Plot of Spearman correlation coefficients for combinations in percentage. A = Physical Activity, Q = QOL, P = Psychic State (Anxiety and Stress), D = Depression, B = Well-Being (Anxiety, Depression and Stress); AQ = Physical Activity X QOL; AP = Physical Activity Influence on Psychic State (Anxiety and Stress); AD = Physical Activity Influence on Depression; AB = Physical Activity Influence on Well-Being (Anxiety, Depression and Stress); QP = QOL Influence on Psychic State (Anxiety and Stress); QD = QOL Influence on Depression; QW = QOL Influence on Well-Being (Anxiety, Depression and Stress).

**Table 1 ijerph-20-01127-t001:** Sociodemographic profile of the sample.

Questions	Answers	Percentage	Absolute
Gender	Feminine	73.6	509
Age-Bracket	60–64	39.4	273
	65–69	32.9	228
Living Status	Family	68.5	474
Marital Status	Married/Stable Union	54.4	376
	Separated/Divorced	22.8	158
Schooling	H.S. Complete	9.2	64
	U.E. Incomplete	9.2	64
	U.E. Complete	38.4	266
	Pos-Graduation	37	257
Physical Activity	Practitioner	84%	582
SRPB	Beliefs give meaning	99%	681
	Feel inner peace	99.6%	687
	Religious	93%	641
	Religious community	79%	545

**Table 2 ijerph-20-01127-t002:** Comparing AS of active and sedentary older people.

AS & AF	AS 2	AS 3	AS 4	AS 5	AS 6
AF0	0 (0.0%)	18 (2.6%)	75 (10.9%)	14 (2%)	1 (0.1%)
AF1	2 (0.3%)	147 (21.3%)	351 (50.9%)	82 (11.9%)	0 (0%)

Caption: AF0 = sedentary; AF1 = active; AS = cluster (anxiety, stress, QOL); AS 2 = null; AS 3 = mild; AS 4 = moderate; AS 5 = moderate; AS 6 = severe.

**Table 3 ijerph-20-01127-t003:** Comparing ADS of active and sedentary older people.

ADS & AF	ADS 3	ADS 4	ADS 5	ADS 6	ADS 7	ADS 8
AF0	0 (0.0%)	16 (2.3%)	58 (8.4%)	26 (3.8%)	6 (9%)	2 (0.3%)
AF1	1 (0.1%)	138 (20%)	297 (43%)	98 (14.2%)	41 (5.9%)	7 (1%)

Caption: AF0 = sedentary; AF1 = active; AS = cluster (anxiety, depression, stress, QOL); ADS 3 = null; ADS 4 = mild; ADS 5 = mild; ADS 6 = moderate; ADS 7 = moderate; AS 8 = severe.

## Data Availability

Not applicable.

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
