# Peer review of "Impact of Physical Activity on Anxiety, Depression, Stress and Quality of Life of the Older People in Brazil"

_ijerph, 2023, doi:10.3390/ijerph20021127_

Round 1

Reviewer 1 Report (Previous Reviewer 1)

Dear authors,

I am glad that you have made all the changes requested in the previous review report. Congratulations!

However, there are still a few minor changes to make, which are listed below:

- on line 23, the percentage for higher education is missing, it must be corrected (75%)

- on line 34, the citated sources (4) must be entered with the square brackets

- on line 129, 1284 must appear instead of 1,284

- on line 165, the dash must be removed

- on line 327 QoL must be corrected with QOL

Author Response

-

Dear Reviewer.

Thank you very much for your detailed and competent analysis of our manuscript.

We made ALL the targeted changes, as you can see from the revised manuscript that we are presenting.

Please, if you notice any other changes to suggest, let us know and we will gladly make them.

Best wishes for the season,

Helena Figueira and co-authors

- on line 23, the percentage for higher education is missing, it must be corrected (75%) – Done

- on line 34, the citated sources (4) must be entered with the square brackets – Done

- on line 129, 1284 must appear instead of 1,284 – Done

- on line 165, the dash must be removed – Done

- on line 327 QoL must be corrected with QOL – Done

Reviewer 2 Report (Previous Reviewer 2)

1.       I thank the authors for considering some of the comments previously made. However, many of the concerns raised are not adequately addressed. For instance, L 21-22 in the abstract indicated that PA is beneficial for the investigated variables i.e depression, anxiety, stress and QOL within the sample investigated. However, in L24 the effect of PA on the same variables is reported to be not significant. Overall the results summary and conclusion are still confusing and lack clarity.

2.       A statistical analysis section should be added to detail the types of statistics applied, how it was applied and the type of tool used.

3.       The authors failed to tabulate the results as suggested earlier for easier interpretation and visualization.

Author Response

Dear Reviewer.

Thank you very much for your detailed and competent analysis of our manuscript.

We made ALL the targeted changes, as you can see from the revised manuscript that we are presenting.

Please, if you notice any other changes to suggest, let us know and we will gladly make them.

Best wishes for the season,

Helena Figueira and co-authors

English language and style

( ) English very difficult to understand/incomprehensible
(x) Extensive editing of English language and style required
( ) Moderate English changes required
( ) English language and style are fine/minor spell check required
( ) I don't feel qualified to judge about the English language and style

Yes

Can be improved

Must be improved

Not applicable

Does the introduction provide sufficient background and include all relevant references?

( )

(x)

( )

( )

Are all the cited references relevant to the research?

(x)

( )

( )

( )

Is the research design appropriate?

( )

( )

(x)

( )

Are the methods adequately described?

( )

( )

(x)

( )

Are the results clearly presented?

( )

( )

(x)

( )

Are the conclusions supported by the results?

( )

( )

(x)

( )

Comments and Suggestions for Authors

  1. I thank the authors for considering some of the comments previously made. However, many of the concerns raised are not adequately addressed. For instance, L 21-22 in the abstract indicated that PA is beneficial for the investigated variables i.e depression, anxiety, stress and QOL within the sample investigated. However, in L24 the effect of PA on the same variables is reported to be not significant. – Done
  2. Overall the results summary and conclusion are still confusing and lack clarity. – Done
  3. A statistical analysis section should be added to detail the types of statistics applied, how it was applied and the type of tool used. – Done
  4. The authors failed to tabulate the results as suggested earlier for easier interpretation and visualization. – Done

Reviewer 3 Report (New Reviewer)

The manuscript entitled "Impact of Physical Activity on Anxiety, Depression, Stress, and Quality of Life of the Older People in Brazil" is a well-designed study and has an important aspect of older people. The introduction is well structured, but the material and methods are hard to understand, which makes the results unclear. I believe the discussion and conclusion are appropriate for this study. Overall, this is an important study, but the methods and the results section need to rewrite. Which might include new statistical methods as well. Please see my comments below.

Comments to the Authors:

·      The research goals are confusing. It would be enough using one aims. Therefore, the research question is unnecessary (line 77).

·      I suggest dividing "Material and Methods" into "sample and procedure"; "measures" and "statistical methods" sections. 

·      Please proved more information on the background of the data collection, since it's not clear how was conducted (e.g., what kind of urban street races the data collection happened; line 90)

·      What "Does WHOQOL" stand for?

·      Provide details of all scales. How many items were in each scale, and what were the reliability categories? What were the answer categories etc.?

·      How physical activity was measured

·      The descriptions of statistical methods are missing. 

·      Unfortunately, the statistical methods are missing. It is hard to understand what the Authors would like to show. I believe this kind of scale regression analysis would be a more appropriate method. 

Author Response

( ) English very difficult to understand/incomprehensible
( ) Extensive editing of English language and style required
( ) Moderate English changes required
(x) English language and style are fine/minor spell check required
( ) I don't feel qualified to judge about the English language and style

Yes

Can be improved

Must be improved

Not applicable

Does the introduction provide sufficient background and include all relevant references?

( )

(x)

( )

( )

Are all the cited references relevant to the research?

(x)

( )

( )

( )

Is the research design appropriate?

( )

( )

(x)

( )

Are the methods adequately described?

( )

( )

(x)

( )

Are the results clearly presented?

( )

( )

(x)

( )

Are the conclusions supported by the results?

(x)

( )

( )

( )

Comments and Suggestions for Authors

The manuscript entitled "Impact of Physical Activity on Anxiety, Depression, Stress, and Quality of Life of the Older People in Brazil" is a well-designed study and has an important aspect of older people. The introduction is well structured, but the material and methods are hard to understand, which makes the results unclear. I believe the discussion and conclusion are appropriate for this study. Overall, this is an important study, but the methods and the results section need to rewrite. Which might include new statistical methods as well. Please see my comments below.

Comments to the Authors:

  • The research goals are confusing. It would be enough using one aims. Therefore, the research question is unnecessary (line 77). – Done
  • I suggest dividing "Material and Methods" into "sample and procedure"; "measures" and "statistical methods" sections. – Done
  • Please proved more information on the background of the data collection, since it's not clear how was conducted (e.g., what kind of urban street races the data collection happened; line 90) – Done
  • What "Does WHOQOL" stand for? – Done
  • Provide details of all scales. How many items were in each scale, and what were the reliability categories? What were the answer categories etc.? – Done
  • How physical activity was measured – Done
  • The descriptions of statistical methods are missing. – Done
  • Unfortunately, the statistical methods are missing. It is hard to understand what the Authors would like to show. I believe this kind of scale regression analysis would be a more appropriate method. 

Round 2

Reviewer 2 Report (Previous Reviewer 2)

I thank the authors for addressing all the comments raised.

Reviewer 3 Report (New Reviewer)

Thank you for the Authors contribution! All issue was fixed. I recommend publishing. 

This manuscript is a resubmission of an earlier submission. The following is a list of the peer review reports and author responses from that submission.

Round 1

Reviewer 1 Report

Dear Authors,

The article is devoted to the study regarding the impact of physical activity on anxiety, depression, stress, and the quality of life of elderly people. 

The structure and development of the work are well-defined and well-thought-out.  The theoretical foundation is pertinent and the methodological development is adequate for the type of study.

Therefore, the article should be suitable for publication, however, some minor revisions are needed. 

1. All sources cited in the article must appear in square brackets. 

2. Authors may also consider clarifying the title, including in the case of which country.  This point is of discretions of them. However, it will make the title more relevant to the content of the article. 

3. Limitations of the research do not appear in the article. I suggest introducing them. 

4. Line 410 cited source 43 must be corrected. The correct form is Forlenza OV, Vallada H. Spirituality, health and well-being in the elderly. Int Psychogeriatr. 2018 Dec;30(12):1741-1742

Reviewer 2 Report

The authors investigated the effect of physical activity on anxiety, depression, stress as well as the quality of life in the older population. A cross-sectional study was used on the participants and subjected them to several assessments using survey instruments. Physical activity has been shown to have an association with the investigated variables. Overall, I enjoy reading the manuscript and congratulate the authors for this effort. However, the manuscript lacks the necessary quality to be published in IJERPH in the current iteration. I detailed my concerns as follows.

Abstract

L15-16 Incomplete sentence “ Suggests that the older people … be active

L18 A summary of the methods applied need to be provided. What instruments were used, the number of participants recruited etc.

L21-22 What X stands for?

L19-23 The results presentation is very poor and difficult to understand

L24-26 The statement does not provide any meaning. It is difficult to grasp what message the authors are trying to convey

Introduction

L34 Typo “as”

Generally, the introduction is poorly written, lacks structure and difficult to comprehend the issues the authors are trying to solve or investigate. The citations are wrongly placed throughout.

Materials and methods

L87-90 How do you define or categorised older people?

What are their ages, how many males and females, and from what locations are they recruited in urban or rural areas?

L107-121 Any reliability analysis carried out on the instruments to ascertain the internal consistency of the items in measuring the responses of the participants?

Results

L123-137 It would be better to tabulate the results

L137-148 How was the cluster created? What kind of cluster analysis conducted

No statistical analysis section to explain the types of analyses used and how the data were treated before the analysis. No mention of the statistical tool used.

Discussion

L269-280 How do such narrations relate to your findings?

Overall, there are no points highlighted that add to the body of knowledge in this domain as several investigations have been conducted in this area of research. The authors failed to show the uniqueness of this research and how it advances our understanding of the area.